# HYBRID NUMERICAL PINNS: ON THE EFFECTIVENESS OF NUMERICAL DIFFERENTIATION FOR COMPLEX PROBLEMS

## ABSTRACT

Automatic differentiation (AD) is the default tool for computing physical derivatives in physics-informed models, but it faces significant limitations when applied to general frameworks, restricting their effectiveness on real-world problems. To overcome these challenges, we propose a hybrid approach that integrates traditional numerical solvers, such as the finite element method, within physics-informed deep learning. This framework enables the exact imposition of Dirichlet boundary conditions seamlessly, and efficiently addresses complex, non-analytic problems. The proposed approach is versatile, making it suitable for integration into any physics-informed model. Crucially, our hybrid gradient computation is up to two orders of magnitude faster than AD, as its computational cost remains unaffected by the underlying model's complexity. We validate the method on representative two and three-dimensional numerical examples and analyze the training dynamics of the hybrid framework.

## 1 INTRODUCTION

Physics-Informed Neural Networks (PINNs) Raissi et al. (2019) have recently emerged as a promising tool to solve Partial Differential Equations (PDEs). These models leverage the expressive power and flexibility of deep learning, making them well-suited for a variety of applications, without requiring costly experimental or numerical data.

Due to their growing popularity, numerous improvements to the physics-informed framework have been proposed. Research has focused on optimizing model architectures, balancing loss terms for forward and inverse problems, and enhancing training procedures. Yet, despite encouraging applications across diverse physical, chemical, and biological problems, challenges remain regarding the convergence of physics-informed models and their application to more complex, non-analytic real-world problems. Specifically, recent works, such as Grossmann et al. (2024), show that physics-informed models struggle to compete with traditional finite element solvers on most benchmarks, both in terms of accuracy and computational efficiency. Furthermore, their reliance on Automatic Differentiation (AD) to compute PDE residuals makes them unable to address non-analytic problems, where, for instance, PDE coefficients are estimated and tabulated instead of given as an analytical field.

In this work, we argue that Automatic Differentiation (AD), commonly used to compute PDE residuals, represents a bottleneck in the current framework. As an alternative, we present hybrid numerical PINNs, where the differential operators are approximated using numerical kernels derived from classical techniques such as the finite element method. We demonstrate that this approach offers significant advantages over Automatic Differentiation-based PINNs. Specifically, the computational cost of this method to calculate the PDE residuals remains constant regardless of the model's complexity, and enables the handling of more general, non analytic problems. Furthermore, Dirichlet boundary conditions can be strongly imposed in a seamless manner.

The paper is organized as follows. First, in Section 2, the physics-informed model and its recent developments are presented. In Section 3, we propose our hybrid numerical PINN, in which the computation of PDE residuals is handled by a numerical gradient kernel, and we highlight the

advantages of this framework over conventional AD PINNs. Then, we apply our hybrid model, first to the Allen-Cahn equation in Section 4, and then to more challenging problems in two and three dimensions in Section 5. The enhancement of the training process is further discussed in Section 6.

## 2 PHYSICS-INFORMED NEURAL NETWORKS

We consider a smooth, open and connex set $\Omega \in \mathbb{R}^d$, with $d \geq 1$. We assume that a function $u$ satisfies a partial-differential equation (PDE) of the following form:

$$\mathcal{N}[x, u] = f(x) \qquad \forall x \in \Omega, \tag{1}$$
$$\mathcal{B}[x, u] = 0 \qquad \forall x \in \partial\Omega. \tag{2}$$

In this formulation, $\mathcal{N}$ and $\mathcal{B}$ are potentially non-linear partial differential and boundary operators and $f$ is a given source term.

In order to approximate the true solution $u$, a neural network with trainable parameters $\theta$ is used to produce a prediction $u_\theta$. *Physics-Informed Neural Networks* Raissi et al. (2019) are designed to solve directly the problem 1-2. To do so, $N_r$ *collocation points* are sampled inside the domain $\Omega$, and $N_b$ points are sampled on the boundary $\partial\Omega$. For more details on the choice of those points, see for instance Wu et al. (2023). These points are used to approximate the PDE residuals inside the domain, therefore the loss term associated to the partial differential operator $\mathcal{N}$ is:

$$\mathcal{L}_r(\theta) = \frac{1}{N_r} \sum_{i=1}^{N_r} \|\mathcal{N}[x_i, u_\theta(x_i)] - f(x_i)\|^2. \tag{3}$$

Similarly, the loss term associated to $\mathcal{B}$ is computed as:

$$\mathcal{L}_b(\theta) = \frac{1}{N_b} \sum_{i=1}^{N_b} \|\mathcal{B}[x_i, u_\theta(x_i)]\|^2. \tag{4}$$

Finally, the total loss function $\mathcal{L}(\theta)$ is the sum of the two loss terms:

$$\mathcal{L}(\theta) = \alpha_r \mathcal{L}_r(\theta) + \alpha_b \mathcal{L}_b(\theta). \tag{5}$$

The weights $\alpha_r$ and $\alpha_b$ are hyperparameters which balance the contribution of the two loss terms during training. For a discussion on the choice of these parameters, see Wang et al. (2021a).

The evaluation of the PDE operator $\mathcal{N}$ is typically made by recording every modification on the input position $x_i$ to produce $u_\theta(x_i)$ inside a *computational graph*, and by deriving every elementary algebraic modification during reverse-mode Automatic Differentiation (AD) Baydin et al. (2018); Margossian (2019). The global PDE operator is then evaluated using the chain rule. These operations are facilitated with the use of deep learning frameworks such as Tensorflow Abadi et al. (2015), Pytorch Paszke et al. (2019) or JAX Frostig et al. (2018).

A discussion on the related works is provided in Annex A.

## 3 PROPOSED APPROACH: NUMERICAL COMPUTATION OF PHYSICAL OPERATORS

### 3.1 NUMERICAL OPERATOR EXTRACTION

In the proposed hybrid numerical physics-informed model, spatial derivative computations are performed using a numerical operator derived from any suitable differentiation technique, such as the

finite difference or finite element method. The key idea is to extract and represent the differential operator as a sparse tensor, enabling its seamless integration into the training framework. This approach offers significant advantages over automatic differentiation-based physics-informed neural networks (AD PINNs), which we discuss in detail in Section 3.2.

Figure 1 illustrates the computation of the loss derivative with the proposed hybrid numerical approach. The extracted operators corresponding to gradient and divergence computations are denoted respectively $\mathcal{G}_\nabla$ and $\mathcal{G}_{\nabla\cdot}$. In terms of computational complexity, once these operators are extracted, computing the gradient or divergence of a field corresponds to a matrix-vector multiplication, therefore it does not require to backpropagate through the computational graph. This operation is therefore independent from the underlying model's complexity.

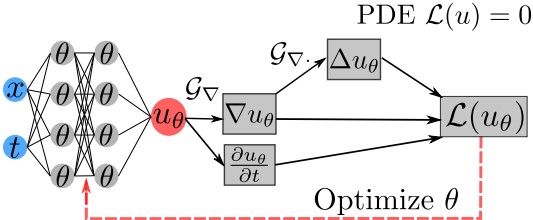

Figure 1: Training workflow of the proposed hybrid physics-informed model. The gradient, divergence and laplacian operators are approximated by sparse operators on a discretized domain, and included inside the training loop. This step is recorded within the automatic differentiation computational graph to facilitate the optimization process.

Our main experiment are presented in Section 5 and are based on a hybrid Finite Element-Physics-Informed approach. To the best of our knowledge, while hybrid Finite Element Physics-Informed models have been proposed before, they typically rely on the full variational finite element formulation of the PDE. In contrast, our method extracts only the derivative operator from the finite element formulation and uses it directly to solve the equation, either in its strong or weak form. While the extraction of a gradient matrix is standard practice in finite difference methods LeVeque (2007), performing a similar operation within a finite element framework is less straightforward. For completeness, we outline the main steps required to construct this finite element gradient operator in Annex B.

### 3.2 NUMERICAL PROPERTIES OF THE HYBRID MODEL

With the extraction of the numerical operator and its inclusion inside the Physics-Informed framework, the resulting model has the following properties:

**Inclusion of non-analytic fields:** Unlike AD, the proposed differentiation operator can be applied to non-analytic fields which are computed outside of the training framework. This capacity allows to address much more general problems, in which the PDE parameters are non-analytic or are derived from closure laws which are not directly implemented. This situation could arise in mechanics for instance, where material parameters appear in the equation but are not known analytically. Instead, they are typically provided in tabular forms. In multiphysics problems, different numerical kernels can solve separate physical laws, and the resulting fields are interchanged to converge to the true, multiphysics solution. All of these settings are beyond the capacities of conventional AD PINNs, as these fields cannot be differentiated, leading to erroneous PDE residuals. In our proposed framework, additional inputs computed by outside solvers can be directly fed to the model without breaking the computational graph, ensuring that the corresponding PDE residuals remain accurate.

**Strong imposition of Dirichlet boundary conditions:** The subject of strongly imposing Dirichlet boundary conditions in physics-informed neural networks is an ongoing research topic Sukumar & Srivastava (2022); Berg & Nyström (2018). Ensuring that the model inherently respects Dirichlet conditions is feasible for general, complex geometries with conventional PINNs. However, this imposition comes with an increased preprocessing complexity (for instance, by fitting distance functions to the boundary). With our hybrid models, this imposition can be done seamlessly, by

directly adjusting the model's predicted values on the boundary to match the Dirichlet values. Suppose the boundary condition in Equation 2 has the following form:

$$u(x) = g(x), \quad x \in \partial\Omega. \tag{6}$$

Given a prediction $u_\theta$ of any physics-informed model, one can strongly enforce such Dirichlet boundary conditions regardless of the geometry with no added complexity using the following operation:

$$u(x) = \mathbb{1}_{\partial\Omega}(x)g(x) + (1 - \mathbb{1}_{\partial\Omega}(x))u_\theta(x). \tag{7}$$

This straightforward procedure cannot be applied to standard AD-based PINNs for the same reasons discussed previously: the differential operator cannot accurately compute derivatives of the solution field near the boundary. As a result, the predicted solution tends to exhibit discontinuities in the vicinity of the boundary. In contrast, numerical differential operators do not require analytical expressions for the imposed Dirichlet conditions, and any discontinuities are naturally penalized during the computation of the PDE residual. We evaluate this procedure on complex geometries with non-analytical Dirichlet conditions in Section 5.

**Constant computation time:** Unlike AD, which records and differentiates every modification of the input to compute the corresponding derivative, the numerical complexity of applying the proposed differential operator is independent of the model. Consequently, for deep neural networks, our method demonstrates significantly faster performance compared to conventional AD, as shown in Section 5, Figure 4. In our experiments, the proposed approach is up to two orders of magnitude faster than AD. Crucially, for more complex architectures such as graph neural networks, which hold great potential for physics-based applications, the speed-up could be even more substantial.

**Better convergence:** Using numerical operators as differential operators simplifies the computational graph of the loss, which, in turn, makes the training smoother, allowing for a better and faster convergence. This aspect is discussed in Section 6.

**Mesh-dependent approach:** Physics-informed models are typically considered meshless approaches, offering a significant advantage over traditional numerical techniques. Our proposed method does not possess this meshless property, as it requires the construction of a mesh to extract the numerical gradient operator. However, our goal is to develop accurate and robust models for two or three-dimensional physics-based simulations in arbitrary geometries, a domain where PINNs still fall short of matching the performance of traditional numerical solvers (see, e.g., Grossmann et al. (2024)). In such contexts, the problem dimensions are typically limited to two or three spatial variables and one temporal dimension, so the curse of dimensionality is not the primary concern. Moreover, while standard PINNs rely on collocation points during training, generating these points becomes increasingly difficult for complex or irregular geometries. Consequently, most of the Physics-Informed models that have tackled such problems rely on meshed versions of the geometry for point sampling Sedykh et al. (2024); Costabal et al. (2024). This suggests that the commonly cited meshless nature of PINNs generally holds for convex and structured geometries. Our aim is to extend the capabilities of physics-informed models to more complex, real-world domains, where incorporating mesh-awareness provides greater flexibility and practical applicability. Therefore, we argue that the need for meshing is not the primary limitation of the proposed method.

## 4 VALIDATION OF THE HYBRID NUMERICAL APPROACH

To compare our hybrid approach to conventional PINNs, the Allen-Cahn equation is chosen, because of its non-linearity. We want to emphasize that for this equation, we did not reach convergence for the AD PINN with a weak imposition of Dirichlet boundary conditions, and therefore used a strong constraint. While for this simple academic benchmark, finding a suitable imposition of such constraint is feasible, more complex problems would require additional preprocessing complexity, with for instance the use of distance functions of other computationally expensive preprocessing

steps Leake & Mortari (2020); Berg & Nyström (2018); Sukumar & Srivastava (2022). The equation solved is the following.

$$\frac{\partial u}{\partial t} = d\frac{\partial^2 u}{\partial x^2} + 5(u - u^3), \qquad (x, t) \in \Omega = (-1, 1) \times (0, 1),$$
$$u(-1, t) = u(1, t) = -1, \qquad\qquad u(x, 0) = x^2 \cos(\pi x). \tag{8}$$

The parameter $d$ is set to $0.001$. This PDE has been solved by finite difference method in Lu et al. (2021), and their provided dataset is used as ground truth. The spatio-temporal domain is discretized as a $201 \times 101$ grid, and the whole grid is used as collocation points.

To strongly impose Dirichlet boundary conditions, The following transformation is implemented for both models. Given the prediction $y_{\text{pred}}$ of the model, the trial solution $u$ is defined as:

$$u(x, t) = x^2 \cos(\pi x) + t(1 - x^2)y_{\text{pred}}(x, t). \tag{9}$$

This transformation, which was also used in the website of Lu et al. (2021), strongly enforces initial and boundary conditions of equation 8. Therefore, the loss function used is only the term associated to the PDE residuals. Two models were trained with the two different derivative computations, and Multi-Layer Perceptrons with three hidden layers of width 64 and the Tanh activation function were selected for both models. The two models were trained for 15,000 epochs with the Adam optimizer and a learning rate of $0.001$, and for 500 additional epochs with the L-BFGS optimizer and a learning rate of $0.5$. Increasing the number of L-BFGS epochs did not decrease the loss. Figure 2 shows the relative $L^2$ error as a function of the training time, for both models.

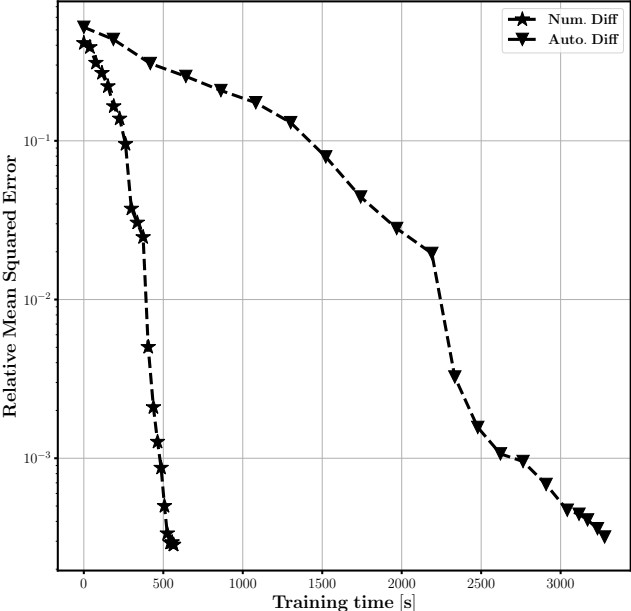

Figure 2: Relative error of the proposed numerical differentiation-based model (stars) and the AutoDiff PINN (triangles), as a function of the training time for the Allen-Cahn equation. The two models reach a similar accuracy, with a speed-up of almost 6 for the hybrid model compared to the conventional AD PINN.

With a relative error of $1.43 \times 10^{-4}$, the proposed model is slightly more accurate than the AD PINN, and its relative error of $1.60 \times 10^{-4}$. However, the main improvement is done regarding the training time, as a speed-up of 5.8 is reached.

## 5 NUMERICAL RESULTS

### 5.1 A TWO-DIMENSIONAL STATIC LINEAR ELASTICITY PROBLEM

To demonstrate the extended ability of numerical PINNs, we extend our analysis to a two-dimensional linear elasticity static problem. The target field in this case is the vector displacement, with Dirichlet boundary conditions on the domain $\Omega$ plotted in Figure 3 and representing the Olympic rings. The target function $\mathbf{u}^* = (u_x^*, u_y^*)$ is obtained with Finite Element Method (FEM). Details on the mathematical formulation of the problem and the training procedure are provided in C.1.

Three models were trained on this problem: one with our hybrid framework with strong imposition of boundary conditions, and two models trained with AD: one with strongly enforced boundary conditions following our approach, and one with a weak imposition of this constraint. The results and the training time are presented in Table 1.

Table 1: Results on the linear elasticity case. The relative Mean Squared Error is reported. 'Hybrid FE-PINN' refers to our hybrid Finite Element (FE) PINN. 'AD PINN, Strong BC' (resp.'AD PINN, Weak BC') refers to the AD PINN, with strongly (resp. weakly) enforced boundary conditions.

| Model | Relative error (%) | Training time (s) |
|---|---|---|
| Hybrid FE-PINN (ours) | **0.05** | $4.97 \times 10^2$ |
| AD PINN, Strong BC | $19,300$ | $1.80 \times 10^3$ |
| AD PINN, Weak BC | $94$ | $1.82 \times 10^3$ |

With a relative mean squared error of $0.05\%$, our model can accurately reconstruct the target solution; the predicted solution is plotted in Figure 3. In contrast, the AD model trained with weak imposition of boundary constraints only achieves a relative error of $94\%$ due to the complexity of the geometry. The performances of the AD method with strongly enforced boundary conditions are even worse, with a relative error of $19,300\%$, demonstrating the difficulty of strongly enforcing boundary conditions with plain PINNs. While more complex architectures like graph models could have better accuracy than plain neural networks, this case highlights the performance of our hybrid approach, and its ability to address real-life geometries and equations.

To further demonstrate the competitiveness of our method in terms of computational complexity, we performed gradient and Laplacian computations using both the traditional AD framework and our finite element-based gradient kernel on the same geometry. The computations have been performed with neural networks of a fixed width of 50, and varying depth (between 1 and 20 hidden layers). For every computation, 10 runs are performed and repeated 25 times for statistical significance. The computations have been performed on a single Nvidia T400 GPU. The average gradient and Laplacian computation times and the associated standard deviations as a function of the neural network's depth are reported in Figure 4.

As expected, the run time of our method does not depend on the model's complexity, making it consistently faster than AD-based computations, especially for more complex models. For the deepest networks, the Laplacian computations are up to two orders of magnitude faster compared to the AD baseline.

The computational graph of our model is also drastically simplified, since no backtracking is needed for the loss computation. This simplification, combined with the results presented in Figure 4, explain the shorter training time of our hybrid method compared to AD PINNs.

### 5.2 HETEROGENEOUS THREE-DIMENSIONAL ELLIPTIC PROBLEM

In this experiment, we consider a non-analytic three-dimensional case on a complex geometry. The problem is an elliptic PDE on an irregular, hollow, three-dimensional coil. The coil is divided into two equal parts, with two materials (steel and Copper Chromium alloy), with different electric conductivities. The target field is the electrostatic potential. Due to the complexity of the geometry, the total conductivity is not an analytical field, therefore, for the reasons developed in Section 3.2, it cannot be differentiated by AD. The details of the problem and the training procedure are provided in Annex C.2. Figure 5 displays the three-dimensional domain, along with the target and predicted field.

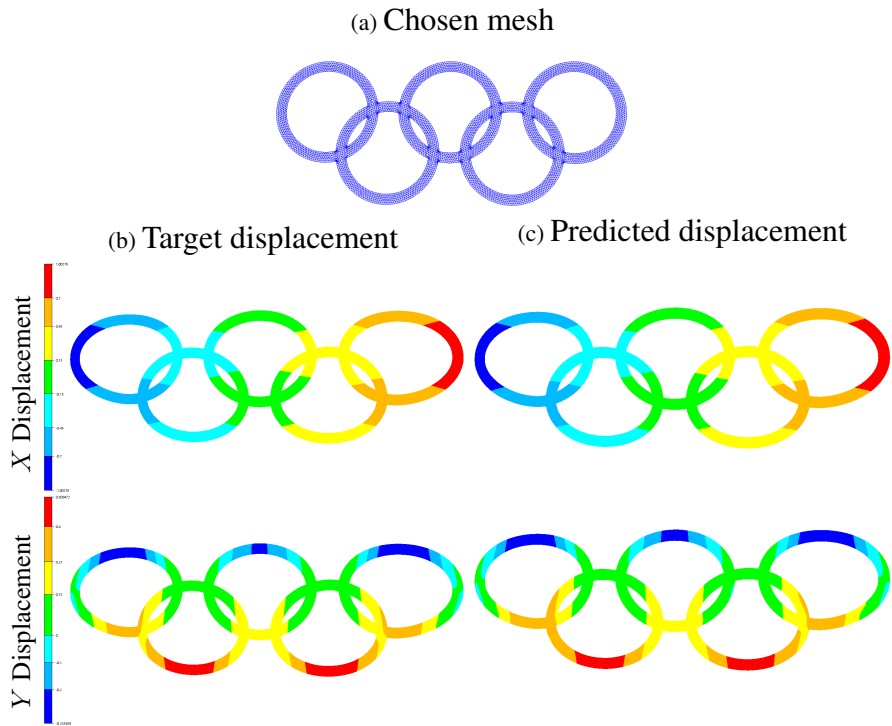

Figure 3: Meshed domain $\Omega$, prediction of our model and target displacement. The boundary nodes where the target $X$ displacement is equal to $-1$ and $1$ are the nodes with Dirichlet boundary conditions.

The results are the following. Our proposed hybrid PINN with strongly enforced boundary conditions achieves a relative error of $1.01\%$, with a training time of $2.04 \times 10^3$ seconds on an Intel Xeon Gold CPU. The conventional PINN, on the other hand, due to its inability to differentiate the conductivity $\sigma(x)$, cannot compute the true PDE residuals. The model reaches a relative error of $42.9\%$, with a longer training time ($9.66 \times 10^3$ seconds). As explained in Section 3.2, such real-life heterogeneous problems seem to be unreachable for conventional PINNs. Meanwhile, our proposed numerical derivation kernel allows the hybrid PINN to converge to the target solution.

## 6    AN INSIGHT ON THE TRAINING OF HYBRID MODELS

The optimization error of a statistical model after the fitting (or training) phase measures how far the model is from the optimal model which minimizes the training loss whithin the same class. Due to the non-convexity of the loss function with respect to their parameters, neural networks are notoriously difficult to train, often resulting in high optimization errors. This is especially verified in physics-informed models, where the target loss function is complex, and involves intricate computational graphs. Specifically, data and PDE loss terms can compete during training, leading to sub-optimal optimizations Wang et al. (2022). In contrast, strongly enforcing Dirichlet boundary conditions mitigates this issue, and the computational graph of numerical differential operators tends to be less complex.

To illustrate this aspect, we computed the loss landscapes of both methods for the models trained to solve the problem presented in Section 5. The loss landscape computation was performed following a straightforward approach inspired by Lorch (2016); Li et al. (2018). During training, the model parameters were stored, and a Principal Component Analysis (PCA) was applied to identify the principal directions in which the parameters evolve. The models' weights were then perturbed along these directions, and the corresponding loss values were recorded. The resulting loss landscapes are presented in Figure 6.

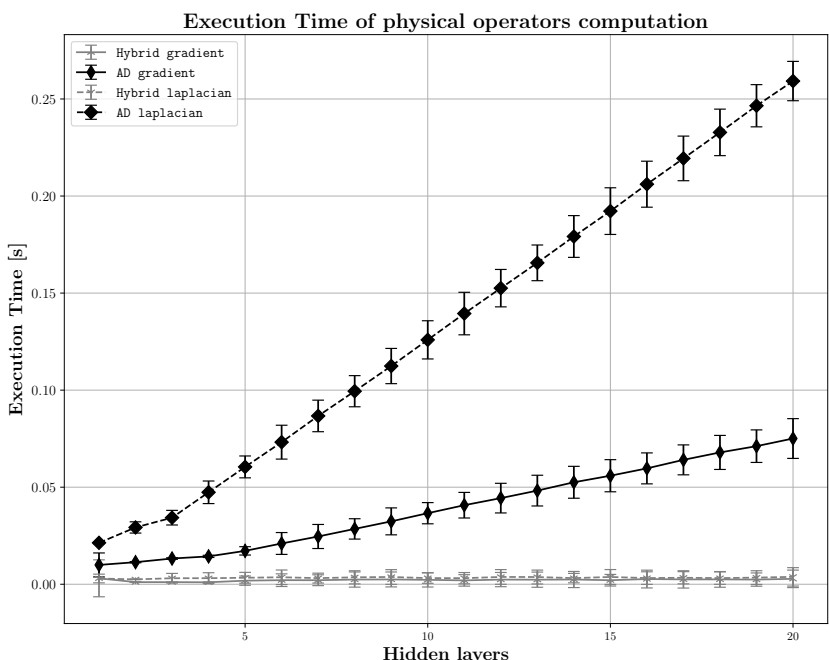

Figure 4: Execution time for gradient (solid lines) and Laplacian (dashed lines) computations in neural networks of varying depth, using both methods. AD operator runtimes are marked with black diamonds, while hybrid computations are indicated by grey chevrons.

While not purely convex, the loss landscape of our hybrid model is noticeably smoother and clearly exhibits a minimum, in contrast to the AD PINN, which has multiple local minima.

The smoothing behaviour of numerical gradients compared to automatic differentiation has already been noticed in other fields such as image rendering or general derivative estimation Petersen et al. (2024); Fischer & Ritschel (2023); Petersen et al. (2022). In our case, this behaviour may be attributed to the competing PDE and boundary loss terms, as well as the increased complexity of the computational graph. The computational graphs of a single gradient computation for both models are presented in Annex D.

## 7 CONCLUSION AND PERSPECTIVES

In this paper, we present hybrid numerical PINNs, and highlight their key improvements over conventional Automatic-Differentiation PINNs. Crucially, our hybrid approach enables the strong imposition of Dirichlet boundary conditions on arbitrary shapes without introducing preprocessing complexity. Additionally, the computation of PDE residuals is decoupled from the model's architecture and complexity, resulting in speed-ups of up to two orders of magnitude. We also discuss and demonstrate the numerical properties of these models, in particular their more stable training processes.

Key future directions include the extension of this setting to more complex problems, such as those involving more challenging boundary conditions and equations, as well as further investigations into the generalization capabilities of such hybrid models. Additionally, expanding our framework to neural operator settings could be of significant interest.

(a) Chosen mesh and subdomain division

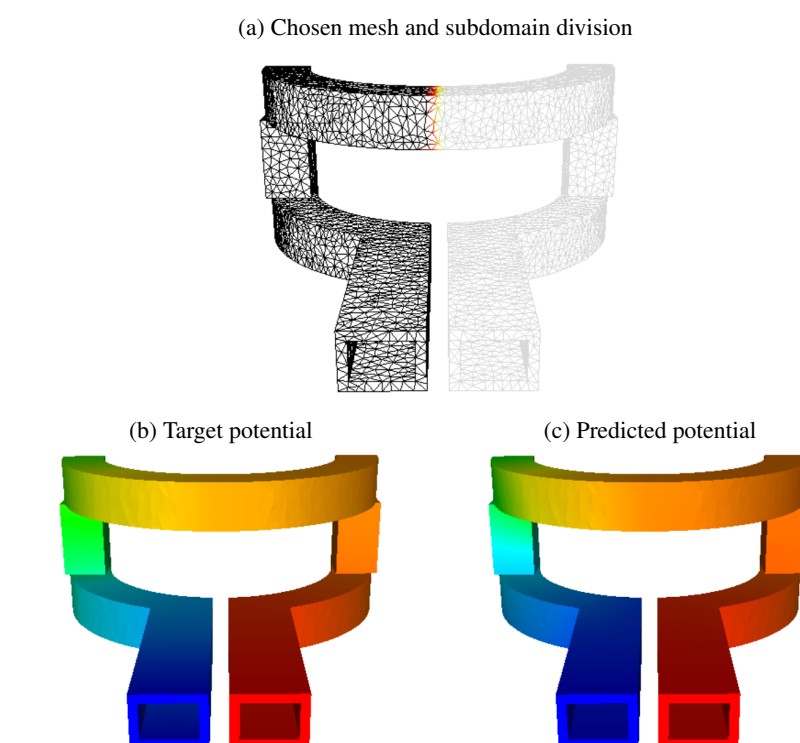

(b) Target potential    (c) Predicted potential

Figure 5: Meshed domain, prediction of our model and target potential. The domain is made of steel (black nodes) and Copper Chromium alloy (grey nodes).

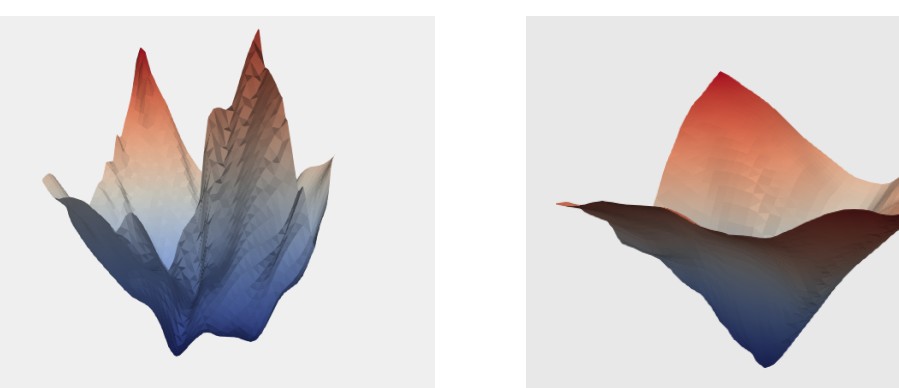

Figure 6: Loss landscapes for AD training (left) and the hybrid framework (right). The loss directions are obtained via PCA on the model's weights during training, with the first two principal directions displayed. The scales of both plots are identical.

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

## A RELATED WORKS

Physics-informed models have proven to be very effective in several academic benchmarks Dissanayake & Phan-Thien (1994); Lagaris et al. (1998a;b); Raissi et al. (2019); Kharazmi et al. (2021). However, deriving theoretical guarantees of convergence for these models is an ongoing research topic Hong et al. (2021); Doumèche et al. (2023). Many efforts have been conducted to enhance the physics-informed framework. Leake & Mortari (2020); van der Meer et al. (2022); Berg & Nyström (2018); Wang et al. (2021a); Sheng & Yang (2022) have addressed the complex issue of balancing the different loss terms accounting for the PDE residuals and the data knowledge (e.g., boundary or initial conditions) to ensure an efficient training. Other approaches have investigated different model architectures Gao et al. (2022); Belbute-Peres et al. (2020); Pfaff et al. (2020); Chenaud et al. (2023; 2024); Dong & Li (2021); Geneva & Zabaras (2020; 2022). For better generalization capabilities, some works have proposed to modify the learning process to fit operators instead of solutions to a single PDE Wang et al. (2021b); Podina et al. (2023); Goswami et al. (2023); Li et al. (2024).

Finally, closer to the approach proposed here, the authors of Belbute-Peres et al. (2020); Meethal et al. (2023); Xiang et al. (2022); Gao et al. (2022) have combined numerical methods with the deep learning approach. In Belbute-Peres et al. (2020); Meethal et al. (2023), the problem of solving the PDE is decomposed between the numerical solver and the deep learning model. In Fang (2021); Xiang et al. (2022); Gao et al. (2022), the residuals are computed without automatic differentiation,

using finite difference methods, and discretized Galerkin analytical formulations. Hybrid Finite Element PINNs are also presented in Costabal et al. (2024); Eshaghi et al. (2024); Meethal et al. (2023). However, to the best of our knowledge, the full finite element formulation is consistently utilized, rather than isolating the differential operator. This design limits the flexibility of the resulting model. Therefore, a quantitative comparison with other hybrid models is challenging, as these approaches typically rely on a fully variational, discretized formulation of the PDE rather than incorporating a numerical differentiation operator directly within the training pipeline. As a result, their implementations are highly problem-specific. Furthermore, to the best of our knowledge, the referenced works do not analyze the improved loss landscapes observed in comparison to AD-based PINNs. Below, we provide a qualitative comparison with similar hybrid approaches.

In Fang (2021), the author introduces a finite difference-based derivative operator that is integrated into the training pipeline, resulting in much shorter training times and improved theoretical convergence guarantees. However, this operator is limited to regular, uniformly spaced grids. The method is extended to irregular meshes through a finite volume formulation of the PDE, which is closer to our hybrid model. However, this approach involves deriving the full finite volume discretization of the equation. As a result, the implementation becomes tightly coupled to the specific PDE under consideration, reducing its adaptability. Moreover, we argue that this formulation is not suitable for general nonlinear equations, where the discretized system matrix depends explicitly on the predicted solution. In such cases, the matrix cannot be precomputed.

In Gao et al. (2022), the problem formulation is integrated within the finite element framework, and, to the best of our understanding, the predicted solution is constrained to lie within a pre-selected FE space. This design choice may limit the model's interpolation capabilities, particularly when applied to a different discretization of the same domain. In contrast, our method uses the mesh solely for computing the derivative operator during training. As a result, the trained model can be evaluated at any point within the domain, independent of the training mesh or discretization.

Finally, the authors of Xiang et al. (2022) propose a radial basis function (RBF) finite difference variant of PINNs. In this approach, the target solution is represented as a combination of RBFs and polynomial basis functions, and the PDE residuals are computed using a discretized version of the governing equation. While this method demonstrates promising results, it is labor-intensive, requiring the construction of two discretized function spaces. Additionally, the formulation is closely tied to the specific PDE being solved, which limits its flexibility.

## B  DERIVATION OF THE FINITE ELEMENT GRADIENT OPERATOR

On a domain $\Omega \subset \mathbb{R}^n$ for which a mesh has been built, we consider a nodal field $u = (u_1, \ldots, u_N)$, $N$ being the number of nodes. For the node $x_i$, $1 \leq i \leq N$, $u_i$ is the value of the field $u$ on the node $x_i$. The $P1$ finite element approximation is based on the consideration of the set of piecewise linear functions $(\varphi_i)_{1 \leq i \leq N}$, such that $\varphi_i(x_j)$ is equal to 1 if $i = j$, and 0 otherwise. The nodal field $u$ is therefore approximated by equation 10.

$$u \approx \sum_i u_i \varphi_i. \tag{10}$$

With this assumption, the spatial gradient of $u$, denoted by $\nabla u$, can be approximated as $\nabla u \approx \mathcal{G}_\nabla u$, with:

$$\mathcal{G}_{\nabla\, i,j} = \frac{1}{\int_{\Omega_i} d\Omega} \int_{\Omega_i} \sum_{g=1}^{n_g} \nabla \varphi_j(x_g) d\Omega. \tag{11}$$

In this formulation, $\Omega_i$ is the set of elements which share the node $i$, $x_g$ are the Gauss points of these elements and $n_g$ is the number of elements. This operation recovers the computation of $\nabla u$ on the Gauss points of the domain, followed by its projection onto the mesh nodes, to recover a $P1$ field.

Numerically, applying this operator is equivalent to a sparse matrix-vector multiplication in terms of computational complexity. Therefore, once the numerical operator $\mathcal{G}_\nabla$ has been extracted from the geometry and converted to a sparse tensor, the numerical gradient computation can be included directly inside any automatic differentiation framework, and its application will be recorded inside the computational graph of the PDE residuals. Similarly, other differential operators such as the divergence and the Laplacian can be obtained as sparse tensors and included in a physics-informed training.

One of the advantages of using such operators is that their behavior is well-understood within the framework of established numerical methods, such as the finite element method (FEM). Theoretical error bounds for these operators have been rigorously studied. For example, the authors of Ern & Guermond (2004) show that, under mild regularity assumptions on the target function, the numerical gradient converges to the true gradient with a rate of $\mathcal{O}(h)$, where $h$ is the mesh size. In our experiments, the numerical error introduced by this approximation was significantly smaller than the optimization error inherent to the PINN, and thus did not constitute the primary limitation in terms of overall accuracy.

## C   DETAILS ON THE NUMERICAL EXPERIMENTS

### C.1   TWO-DIMENSIONAL LINEAR STATIC ELASTICITY PROBLEM

The mathematical formulation of the problem is the following:

$$
\begin{aligned}
\operatorname{div} \boldsymbol{\sigma}(\varepsilon) &= \mathbf{0}, \\
\varepsilon(\mathbf{u}) &= \nabla \mathbf{u} + \nabla \mathbf{u}^T, & \boldsymbol{\sigma}(\varepsilon) &= \lambda \operatorname{Tr}(\varepsilon)\mathbf{I} + 2\mu\varepsilon, \\
\mathbf{u}(x,y) &= \mathbf{u}^*(x,y), & (x,y) &\in \Gamma \subset \partial\Omega, \\
\frac{\partial \mathbf{u}}{\partial \mathbf{n}}(x,y) &= 0, & (x,y) &\in \partial\Omega \backslash \Gamma.
\end{aligned}
\tag{12}
$$

The Lamé parameters $\lambda$ and $\mu$ are set to 1, and $\mathbf{I}$ denotes the identity matrix. A mesh of 5,104 nodes was built on this complex geometry. The FEM computation took 0.21 second and 1,452 iterations for a relative tolerance of $10^{-4}$, and 0.4 second (2,843 iterations) for a relative tolerance of $10^{-8}$ using a preconditioned conjugate gradient method with Jacobi preconditioner. The experiments were made on a single Intel Xeon Gold CPU. The FEM gradient kernel has been used for our hybrid approach, following the approach presented in Section 3.

For the three models tested, the training has been conducted for 20,000 epochs with the Adam optimizer, and a learning rate of 0.005. The models are Multi-Layer Perceptrons with 3 hidden layers of width 50, and with the Tanh activation function. To include the homogeneous Neumann boundary condition, a weak formulation of the PDE is used to compute the residuals.

### C.2   THREE-DIMENSIONAL ELLIPTIC PROBLEM

The target field is the electrostatic potential $V$. Its value is set to zero on one extremity of the coil, $\partial\Omega_1$, and to 1 on the other extremity, $\partial\Omega_2$. The full mathematical formulation is the following:

$$
\begin{aligned}
\nabla \cdot (\sigma(x)\nabla V(x)) &= 0, & x &\in \Omega, \\
V(x) &= 0, & x &\in \partial\Omega_1, \\
V(x) &= 1, & x &\in \partial\Omega_2.
\end{aligned}
\tag{13}
$$

The domain is divided into two parts, with two different materials. The conductivity $\sigma$ is equal to $2.225 \times 10^7$ for the Copper Chromium alloy part, and to $6.25 \times 10^6$ for the steel part. The domain is discretized into a mesh composed of 10,368 nodes. This problem has been solved by finite element method, and the corresponding result is considered to be the ground truth. Two models are trained on this problem: our proposed hybrid PINN with the numerical gradient kernel, and a strong imposition of the Dirichlet boundary conditions, and a conventional AD PINN with weakly enforced boundary conditions. Both models are multi-layer perceptrons with three hidden layers of width 100, and the

Tanh activation function. The training is conducted for $5,000$ epochs with the Adam optimizer and a learning rate of $0.001$, and for $1,000$ additional epochs with the L-BFGS optimizer, and a learning rate of $0.1$. The loss is the $L^1$ norm of the physics-informed residuals for our hybrid model, and the sum of the physics-informed loss term and the Dirichlet boundary error for the conventional PINN.

## D  A COMPARISON OF THE COMPUTATIONAL GRAPHS

To demonstrate the ability of our method to simplify the computational graph of a physics-informed loss, we computed the gradient of two neural networks with the same architecture (3 hidden layers of width 50, and the Tanh activation function), on the geometry presented in Section 5.1. We captured the computational graph of this operation, and visualized it. The results are presented in Figures 7 and 8.

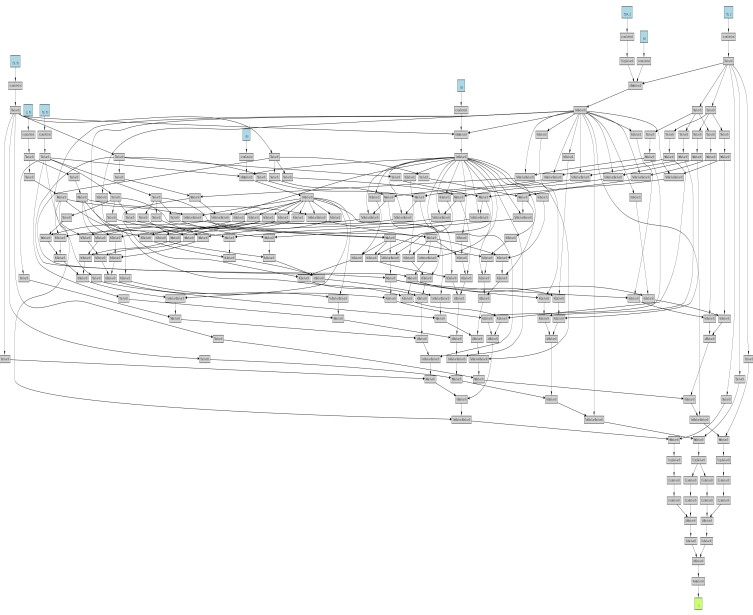

Figure 7: Computational graph of the AD gradient computation.

Unlike the AD operator, the hybrid differential operator does not require performing backward passes on the computational graph to complete the forward computation, which explains the significantly simpler structure of the corresponding graph. This property may also explain the smoother loss landscape observed in Section 6.

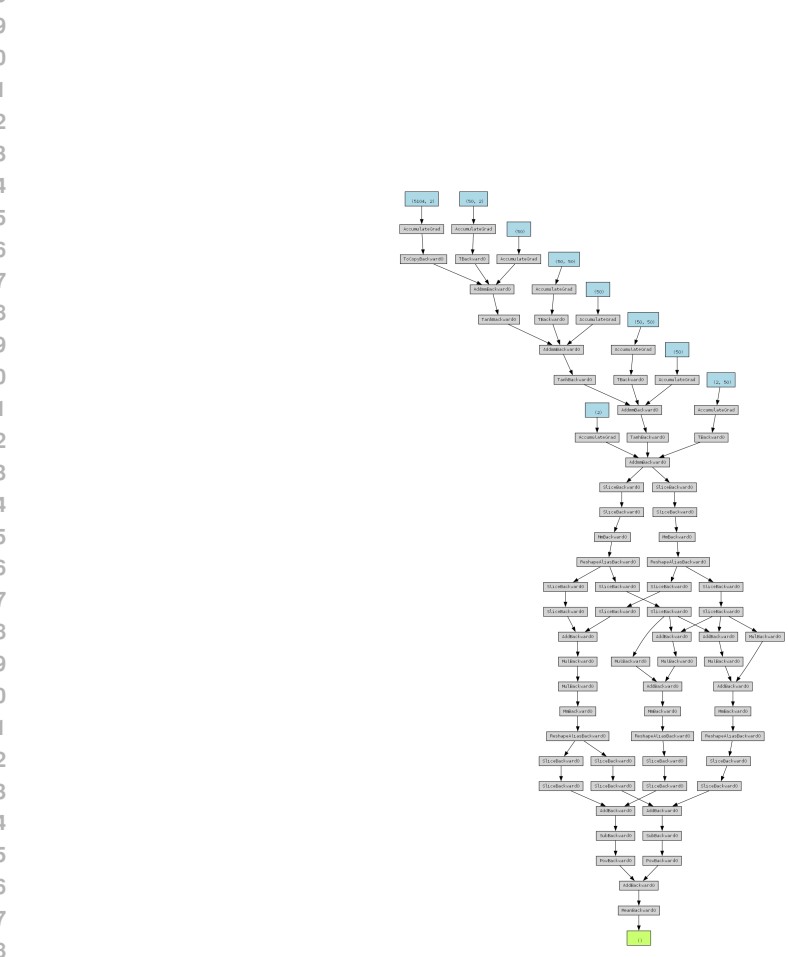

Figure 8: Computational graph of the hybrid gradient computation.

