# OpenReview forum: "Hybrid Numerical PINNs: On the effectiveness of numerical differentiation for complex problems"
_ICLR.cc/2026/Conference — ICLR 2026 Conference Withdrawn Submission_

### Official Review · Reviewer_8m7f · 2025-10-21

**Soundness:** 1
**Presentation:** 1
**Contribution:** 2
**Rating:** 2
**Confidence:** 2

**Summary:**

The authors replace automatic differentiation (AD) -based PDE residuals in PINNs with numerical differentiation (sparse operators extracted from FEM, etc.). Training becomes faster and more robust with costs that are independent of model complexity. The key idea is to extract differential operators as sparse tensors, enabling its seamless integration into the training loop.

On Allen–Cahn (1D+time), the method matches AD PINN accuracy while being ≈5× faster; on 2D linear elasticity and a 3D multi-material coil, it outperforms AD especially when coefficients are non-analytic.

Additionally, numerical differentiation simplifies the computation graph and appears to smooth the loss landscape; visualizations suggest more stable training.

**Strengths:**

- (Technical contribution) The computational cost of the proposed method to calculate the PDE residuals remains constant regardless of the model’s complexity and can handle non-analytic problems.

- (Clarity) Related work on AD, numerical differentiation, and PINNs are thoroughly discussed in Appendix A, highlighting the contribution of the paper.

- (Reproducibility) Well-defined error bars are presented.

**Weaknesses:**

- (Major comment: conceptual flaw) The proposed method is designed for PINNs but relies on a mesh, thereby undermining one of PINNs’ key advantages (i.e., meshless) over conventional numerical solvers.

- (Major comment: reproducibility) The code is not provided, and implementation details are unclear (though some are described in Appendix C), which undermines the reproducibility of the work. Could the authors make it available or explain the reason for its unavailability?

- (Clarity) The main text is not self-contained: the explanation of the proposed method (lines 105-139) heavily relies on Appendix B and is hard to follow. Does it simply replace AD with a conventional numerical differentiation method? How is it implemented into the standard PyTorch training loop?

- (Empirical evaluation) The paper would benefit from discussing mesh size dependence. How can we avoid numerical errors of the standard numerical methods such as FEM?

- (Empirical evaluation) In Table 1, 19,300% error for AD (strong BC) appears pathological (and even 94% (weak BC) is very high). Please check metric definitions, unit scaling, and normalization, or provide a reference that indicates this is normal.

- (Minor comment) Figure 7 is too small to read clearly and appears pixelated.

### Review Summary

The paper introduces a hybrid method that accelerates PINN training. However, the proposed approach is difficult to follow, making its advantages hard to assess. Moreover, I am concerned about the conceptual flaw and limited reproducibility. I am therefore inclined to recommend rejection.

**Questions:**

See also Weaknesses above.

---

### Official Review · Reviewer_s8Wo · 2025-10-26

**Soundness:** 2
**Presentation:** 2
**Contribution:** 2
**Rating:** 2
**Confidence:** 5

**Summary:**

This paper proposes a Hybrid Numerical PINN framework that replaces automatic differentiation (AD) with numerical differentiation kernels—such as finite element–derived gradient operators—to compute PDE residuals during training. The authors claim that this modification yields faster computation, allows handling of non-analytic coefficients, and enables strong imposition of Dirichlet boundary conditions. Experiments include (1) the 1D Allen–Cahn equation, (2) a 2D static linear elasticity problem, and (3) a 3D elliptic problem.

**Strengths:**

**Strengths**
- The paper is clearly written and easy to follow.
- The integration of numerical differentiation (e.g., FEM-based operators) into the PINN framework is technically straightforward and may offer practical benefits.
- The authors present test cases in 1D, 2D, and 3D, showing some computational speed-up compared to AD-based PINNs.

**Weaknesses:**

**Weaknesses**
- **Lack of novelty:** The central idea—replacing automatic differentiation with numerical derivatives—is a basic and well-known concept, not a novel research contribution. Similar “hybrid” or “numerical PINN” ideas already exist (e.g., arxiv.org/pdf/2205.08321, arxiv.org/pdf/2409.02810).
- **Limited experiments:** The examples (Allen–Cahn, 2D elasticity, 3D elliptic problem) are too simple. More diverse PDEs and geometries are needed to demonstrate generality and robustness.
- **Outdated comparison:** The study only compares with the original PINN. Recent advancements (e.g., arXiv:2402.00326, arXiv:2306.15969, and many others) are completely ignored, making the experimental evaluation obsolete and unconvincing.
- **No theoretical foundation:** The paper lacks mathematical justification, convergence analysis, or error estimates for the proposed hybrid scheme.
- **Contradicts the purpose of PINNs:** The method requires meshing to compute numerical derivatives, eliminating the key advantage of PINNs as mesh-free solvers. Without mesh-free capability, it becomes unclear why PINNs are needed instead of traditional FEM/FVM.
- **Missing discussion on boundary conditions:** Only Dirichlet conditions are considered. The approach for inhomogeneous Neumann or Robin boundaries is not discussed.
- **Overstated claims:** The claimed “two orders of magnitude” speed-up and “better convergence” are shown only on toy examples, without rigorous ablation or scalability studies.

Overall, the paper reads like an engineering implementation note rather than a research-level study, with minimal novelty, shallow analysis, and outdated benchmarking.

**Questions:**

The main questions for the authors are already reflected in the **Weaknesses** section above, as they directly arise from the identified issues.

---

### Official Review · Reviewer_foyH · 2025-10-28

**Soundness:** 2
**Presentation:** 2
**Contribution:** 1
**Rating:** 2
**Confidence:** 4

**Summary:**

The paper considers the problem of learning PDE solutions with PINNs, and proposes Hybrid Numerical PINNs. The method leverages meshed-based numerical solvers to get PDE operators in prior, to improve gradient computation. The author demonstrated computation and prediction accuracy improvements over automatic differentiation based PINNs, on 3 problems.

**Strengths:**

1. The paper is in general clearly written and easy to follow.
2. The authors demonstrated reduced gradient computation over automatic differentiations, assuming a mesh is given in prior.

**Weaknesses:**

1. The idea of combining numerical solvers such as FEM with PINN [1-3], or direct imposition of boundary conditions [4-6] in PINN are not new. There needs to be more detailed comparison with those prior works both conceptually and experimentally.
2. The experiment is far from comprehensive. Only fixed PDE without varying parameters are considered, and there misses benchmark problems such as ones in [7-8].
3. The method seems to lack practical usefulness. The advantage of PINN is to solve PDEs with varying parameters and for cases where meshes could be hard to obtain. If a mesh is given, why would one use the proposed method over numerical solvers?

[1] Sobh, Nahil, Rini Jasmine Gladstone, and Hadi Meidani. "PINN-FEM: A Hybrid Approach for Enforcing Dirichlet Boundary Conditions in Physics-Informed Neural Networks." arXiv preprint arXiv:2501.07765 (2025).

[2] Feng, Xiaodong, et al. "A hybrid FEM-PINN method for time-dependent partial differential equations." arXiv preprint arXiv:2409.02810 (2024).

[3] Zhang, Ning, et al. "Finite element-integrated neural network framework for elastic and elastoplastic solids." Computer Methods in Applied Mechanics and Engineering 433 (2025): 117474.

[4] Wang, Zhuoyuan, et al. "Physics-Informed Deep B-Spline Networks for Dynamical Systems." arXiv preprint arXiv:2503.16777 (2025).

[5] Liu, Songming, et al. "A unified hard-constraint framework for solving geometrically complex pdes." Advances in Neural Information Processing Systems 35 (2022): 20287-20299.

[6] Berrone, Stefano, et al. "Enforcing Dirichlet boundary conditions in physics-informed neural networks and variational physics-informed neural networks." Heliyon 9.8 (2023).

[7] Takamoto, Makoto, et al. "Pdebench: An extensive benchmark for scientific machine learning." Advances in Neural Information Processing Systems 35 (2022): 1596-1611.

[8] Zhongkai, Hao, et al. "Pinnacle: A comprehensive benchmark of physics-informed neural networks for solving pdes." Advances in Neural Information Processing Systems 37 (2024): 76721-76774.

**Questions:**

1. How does the proposed method compare with [1-6]?
2. How does the proposed method perform under PDEs problems considered in [7-8]?
3. In figure 2, the prediction error drops faster with numerical differentiation, but in figure 6 the optimization landscape of automatic differentiation seems sharper. Why is this discrepancy?

---

### Official Review · Reviewer_yhPV · 2025-10-31

**Soundness:** 2
**Presentation:** 2
**Contribution:** 2
**Rating:** 2
**Confidence:** 4

**Summary:**

This paper replaces automatic differentiation in PINNs with numerical differential operators that are applied as fixed linear kernels inside training. This lets model impose Dirichlet BCs effectively to address complex problems. Authors showcase that the method is faster than AD-based implementations. Experiments showcase improvements in efficiency and accuracy.

**Strengths:**

- clear, practical, and simple idea.
- replacing AD with numerical techniques is a reasonable and exciting direction for PINNs.
- concrete benefits in terms of speedups and accuracy over AD based PINNs.

**Weaknesses:**

- extremely thin baseline and PDE selections: the experimental framework is lacking. I would recommend benchmarking against some SOTA PINN variants such as: PINNMamba, PINNsFormer, and RoPINNs on the PINNacle benchmark. This is currently a standard practice.

- differentials obtained from AD should be more accurate than the numerical techniques. There is currently no discussion in the paper on this.

- The approach forfeits the classic “meshless” appeal of PINNs. Authors argue this is acceptable for 2D/3D real geometries, but it’s still a trade-off.

- PDEs used in experiments are steady elliptic and Allen-Cahn; no time-dependent 2D/3D with complex BCs, shocks, or advection.

- The headline “orders-of-magnitude” speedups are operator timings. end-to-end training speedups are smaller and measured on mixed hardware (GPU for micro-benchmarks; CPU for 3D coil). A unified wall-clock study would help.

**Questions:**

- Can the authors add modern SOTA AD-PINN baselines and the PINNacle benchmark?

- Can the authors provide a unified wall-clock study, and report memory?

---

### Note · Authors · 2025-12-01

I have read and agree with the venue's withdrawal policy on behalf of myself and my co-authors.